# Intranasal type I interferon treatment is beneficial only when administered before clinical signs onset in the SARS-CoV-2 hamster model

**Pierre Bessière**[1], **Marine Wasniewski**[2☯], **Evelyne Picard-Meyer**[2☯], **Alexandre Servat**[2☯], **Thomas Figueroa**[1], **Charlotte Foret-Lucas**[1], **Amelia Coggon**[1], **Sandrine Lesellier**[3], **Frank Boué**[2], **Nathan Cebron**[1], **Blandine Gausserès**[1], **Catherine Trumel**[4], **Gilles Foucras**[1], **Francisco J. Salguero**[5], **Elodie Monchatre-Leroy**[2], **Romain Volmer**[1]*

1 Ecole nationale vétérinaire de Toulouse, ENVT, INRAE, UMR 1225, IHAP, Université de Toulouse, Toulouse, France, 2 Nancy laboratory for rabies and wildlife, ANSES, Lyssavirus Unit, Malzéville, France, 3 Nancy laboratory for rabies and wildlife, ANSES, Atton experimental facility, Atton, France, 4 Ecole nationale vétérinaire de Toulouse, ENVT, CREFRE, INSERM, Université de Toulouse, Toulouse, France, 5 National Infection Service, Public Health England (PHE), Porton Down, Salisbury, Wiltshire, United Kingdom

☯ These authors contributed equally to this work.
* romain.volmer@envt.fr

**Data Availability Statement:** All relevant data are within the manuscript and its Supporting Information files.

## Abstract

Impaired type I interferons (IFNs) production or signaling have been associated with severe COVID-19, further promoting the evaluation of recombinant type I IFNs as therapeutics against SARS-CoV-2 infection. In the Syrian hamster model, we show that intranasal administration of IFN-α starting one day pre-infection or one day post-infection limited weight loss and decreased viral lung titers. By contrast, intranasal administration of IFN-α starting at the onset of symptoms three days post-infection had no impact on the clinical course of SARS-CoV-2 infection. Our results provide evidence that early type I IFN treatment is beneficial, while late interventions are ineffective, although not associated with signs of enhanced disease.

## Author summary

Type I interferons are major antiviral effectors produced by the host in response to viral infections. Importantly, delayed or impaired type I IFN signalling response has been shown to correlate with severe COVID-19. These observations provided further impetus to test the administration of exogenous type I IFN as a treatment against SARS-CoV-2 infection in patients. However, studies using MERS-CoV or SARS-CoV infected mice demonstrated that type I interferon treatment was beneficial when administered early, but was ineffective and even caused deleterious immunopathology when administered at later stages of infection. It is therefore crucial to understand how the timing of the type I IFN

**Funding:** This work was funded by a grant from the Agence Nationale de la Recherche (ANR-20-COV5-0004) to RV. The funders had no role in study design, data collection and analysis, decision to publish, or preparation of the manuscript.

**Competing interests:** The authors have declared that no competing interests exist.

treatments modulates their efficacy and safety against SARS-CoV-2. In this preclinical study using the SARS-CoV-2-infected Syrian hamster model, we showed that intranasal type I IFN treatment was beneficial only when administered before the onset of symptoms. Importantly, late treatment was ineffective but was not associated with deleterious effects. This study provides important information to interpret clinical trials showing no to modest effects of type I IFNs in COVID-19 patients.

## Introduction

Type I interferons (IFNs) are major antiviral cytokines and their finely tuned production is critical for host protection against viruses [1]. *In vitro* studies demonstrated that SARS-CoV-2 was very sensitive to the antiviral effects of type I IFN [2–4]. In addition, development of severe COVID-19 was shown to correlate with decreased type I IFNs production or impaired type I IFN signaling [5–9]. In agreement with these observations, recombinant type I IFNs are being tested in a number of clinical trials to treat COVID-19 patients [10–12]. However, the design and interpretation of these clinical trials need to consider that the timing of type I IFN treatment may be critical for its efficacy and safety against SARS-CoV-2 [13,14]. Indeed, studies in SARS-CoV and MERS-CoV infected mice demonstrated that type I IFN-treatment was beneficial when administered early, while it was deleterious when administered at later stages of infection [15,16]. How the timing of type I IFN treatment modulates clinical efficacy against SARS-CoV-2 is currently unknown and needs to be tested in an animal model.

## Results

To address this question, we first determined the consequences of recombinant universal IFN-α (IFN) (Hu-IFN-αA/D[Bg/II], pbl assay science, Piscataway, NJ) on the expression of the type I IFN stimulated gene (ISG) Mx1. Mx1 is an ISG and its level of expression is a good indicator of the levels of type I and type III IFN acting locally [17]. We observed a significant upregulation of Mx1 expression in the nasal turbinates, lungs and spleen of hamsters treated intranasally with $10^5$ IU IFN, demonstrating that this molecule was active in hamsters (Fig 1A). Pulmonary Mx1 gene expression 24 hours post IFN treatment did not differ significantly between animals treated with $10^5$ IU IFN or with $7.10^5$ IU IFN (Fig 1B). At 48 hours post treatment with $10^5$ IU IFN, pulmonary Mx1 mRNA expression was reduced compared to 24 hours post treatment with the same dose, but remained upregulated compared to placebo treatment (Fig 1B). Next, we analyzed Mx1 protein expression in the lungs of IFN-treated hamsters by immunohistochemistry. In IFN-treated hamsters, Mx1 protein expression was detected in the main target cells of SARS-CoV-2, including pneumocytes, bronchiolar and bronchial epithelial cells, but also in endothelial cells and immune cells within the lung parenchyma (Fig 1C). The percentage of Mx1 positive lungs was significantly increased 24 hours post-treatment in animals administered $10^5$ IU IFN and further increased in animals administered $7.10^5$ IU IFN (Fig 1D). Importantly, in animals administered $10^5$ IU IFN, Mx1 positive lung area was equivalent at 24 hours and 48 hours post-treatment, indicating that Mx1 protein levels remained elevated for 48 hours following type I IFN intranasal administration (Fig 1D), in accordance with a previous report [18]. We thus decided to treat hamsters every two days in an effort to minimize the side effects due to the anesthesia required to treat hamsters intranasally with IFN. In human clinical trials, nebulized type I IFNs are being tested at $6.10^6$ IU per treatment, which corresponds to a hamster equivalent dose of approximately $10^5$ IU per hamster based on body

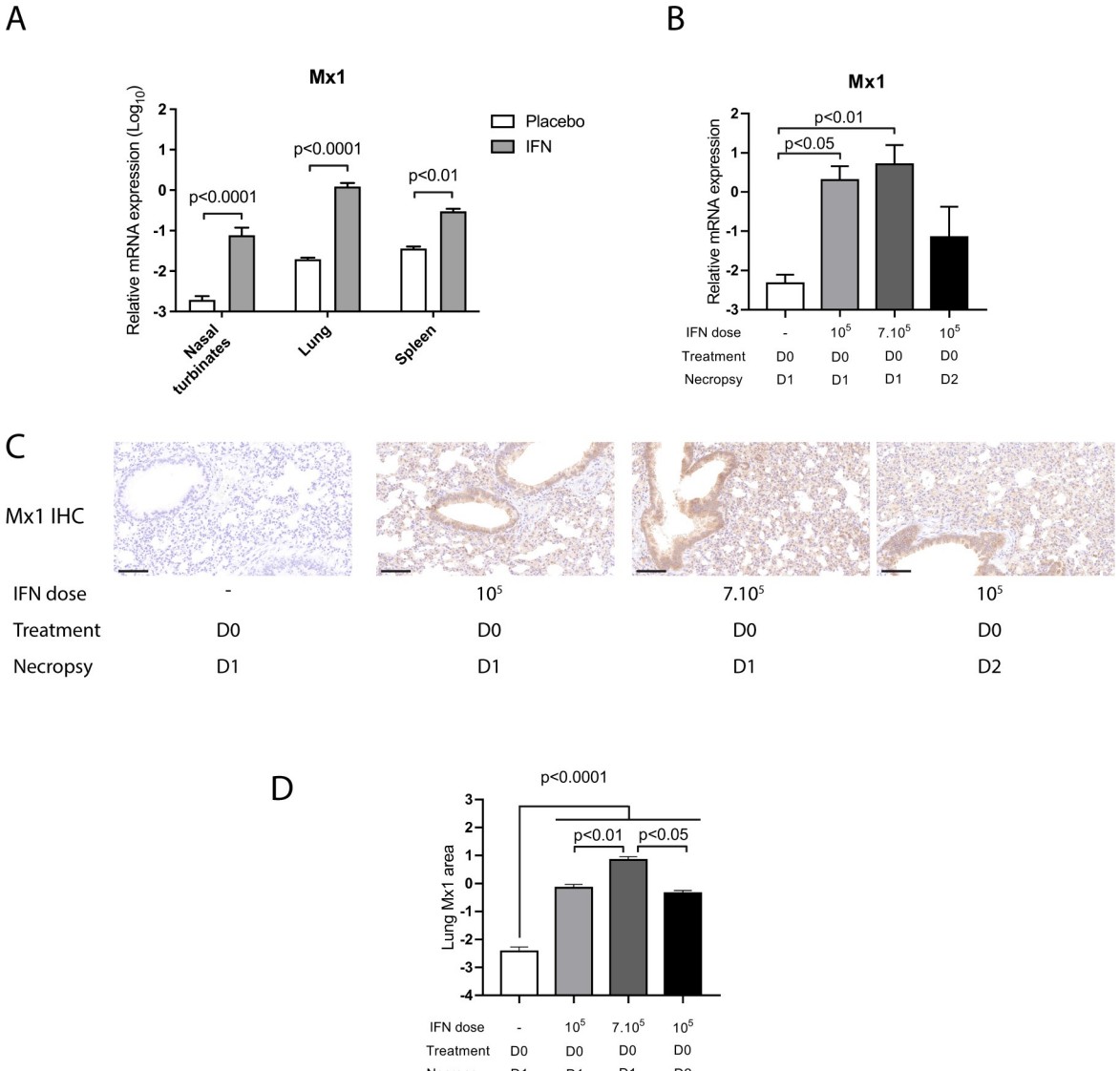

**Fig 1. Impact of IFN-α treatment on Mx1 lung transcript and protein levels in non-infected hamsters. (A)** Syrian hamsters were treated intranasally either with placebo or with $10^5$ IU recombinant universal IFN-α (IFN). Tissues were harvested at day 1 post-treatment. Transcripts levels of Mx1 relative to the housekeeping genes RPL18 and RPS6KB1 were determined by RT-qPCR. Results are expressed as means ± SEM. Statistical analysis: one-way ANOVA with Tukey's multiple comparisons test. **(B-D)** Syrian hamsters were treated intranasally either with placebo or with $10^5$ UI IFN or with $7.10^5$ IU IFN. Tissues were harvested either at day 1 or day 2 post-treatment. **(B)** Lung transcripts levels of Mx1 relative to the housekeeping genes RPL18 and RPS6KB1 determined by RT-qPCR. **(C)** Representative pictures were selected to display Mx1 lung protein levels detected by immuno-chemistry (IHC). Scale bar: 100μm. **(D)** Quantification of percent lung area positive for Mx1 protein detected by IHC. D0: day 0; D1: day 1; D2; day 2. Statistical analysis: one-way ANOVA with Tukey's multiple comparisons test.

surface area conversion, as described in *Materials and Methods* [12,19]. We therefore treated hamsters with $10^5$ IU IFN per hamster in the following experiments.

Next, we designed a study that evaluated the prophylactic and therapeutic efficacy of intranasally administered recombinant IFN against SARS-CoV-2 infection in Syrian hamsters (Fig 2A). Hamsters intranasally infected with a high SARS-CoV-2 dose develop clinical disease caused by lung pathology, which closely mirrors severe human COVID-19 [20,21]. Following challenge with $10^4$ TCID$_{50}$ SARS-CoV-2, we observed significant weight loss in the placebo-

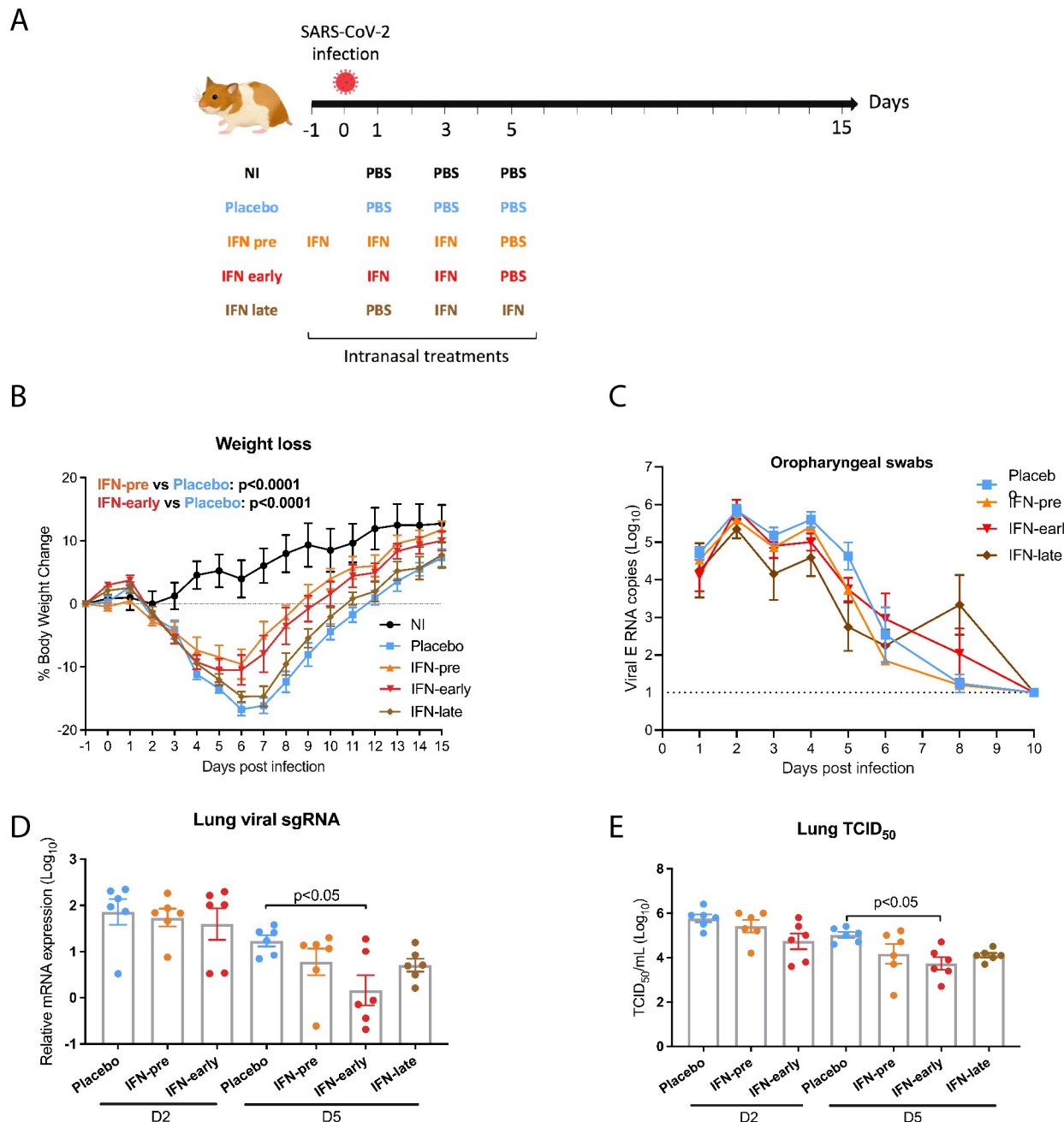

**Fig 2. Impact of type I IFN-α treatment on SARS-CoV-2-induced weight loss and viral titers. (A)** Overview study design. **(B)** Percentage of body weight change with weight measured day 1 pre-infection before the IFN-pre treatment set as the reference weight (6 animals per group). Statistical analysis: two-way ANOVA comparing treatment effects with Geisser-Greenhouse correction followed by Tukey's multiple comparisons test. **(C)**. Viral genomic RNA in oropharyngeal swabs (6 animals per group). The dotted line indicates limit of detection. **(D and E)** Lungs viral titers determined by RT-qPCR targeting viral sgRNA relative to the housekeeping genes RPL18 and RPS6KB1 **(D)** and by $TCID_{50}$ **(E)**. D2: day 2 post infection; D5: day 5 post infection. Results are expressed as means ± SEM. Statistical analysis: one-way ANOVA with Tukey's multiple comparisons test.

treated animals, as expected with a high SARS-CoV-2 inoculum titer [20,21]. No protection from weight loss was observed in the IFN-late group, for which treatment was initiated at the onset of clinical signs, when infected animals started to significantly lose weight three days

post-infection (Fig 2B). By contrast, we observed a significant protection from weight loss in the IFN-pre group (prophylactic treatment initiated 16 hours before infection) and in the IFN-early group (treatment initiated at one day post-infection) compared to the placebo group (Fig 2B). The protection from weight loss in the IFN-pre and in the IFN-early groups was not associated with a reduction of viral excretion level or duration, as viral RNA levels measured by RT-qPCR from oropharyngeal swabs were similar in all groups (Fig 2C). In agreement with this observation, subgenomic viral RNA levels in the nasal turbinates were similar in all groups (S1 Fig). As SARS-CoV-2 respiratory disease is due to lower respiratory tract damage, we analyzed viral load in the lungs. We detected a reduction of pulmonary viral subgenomic RNA levels and infectious viral titers in all the IFN-treated groups at day 5 post-infection, compared to the placebo group, which reached statistical significance in the IFN-early group only (Fig 2D and 2E).

Evaluation of the respiratory tract from infected animals revealed a mild to moderate bronchointerstitial pneumonia at day 2 post-infection, progressing to moderate/severe with lung consolidation at day 5 and resolving at day 15 with only small lesioned areas remaining, as previously observed [21]. The lesions were characterized by infiltrates of macrophages and neutrophils, with fewer lymphocytes and plasma cells (Figs 3A and S2). A reduction of the lung pathology scores was observed in the IFN-treated groups compared to the placebo group, which reached statistical significance in the IFN-early group only (Fig 3B). RNAScope *in situ* hybridization (ISH) was used to determine the localization of viral RNA in the lungs of infected animals. Viral RNA was observed in bronchial and bronchiolar epithelial cells and in regions of inflammatory infiltrates at day 2 post-infection (S2 Fig). The viral RNA positive area diminished at day 5 and coincided with inflammatory infiltrates. Quantification of viral RNA positive area revealed a slight non-statistically significant reduction of viral RNA in the IFN-pre and in the IFN-early groups at day 2 and 5 post-infection compared to the placebo group (Fig 3C). Mx1 protein was upregulated in the lungs of infected hamsters, as detected by immunohistochemistry, and the percentage of Mx1 positive lung was equivalent in placebo and IFN-treated hamsters (Figs 3D and S2). Finally, hematological analyses revealed a modest lymphocytopenia in SARS-CoV-2 infected hamsters, with no difference between the IFN-treated groups and the placebo group (S3 Fig).

To explore the consequences of type I IFN administration on the immune response to SARS-CoV-2 infection, we analyzed the gene expression of immune markers gene expression from the lungs of animals euthanized at day 2 and 5 post-infection. Compared to the non-infected animals, all the infected groups presented a significant upregulation of the ISGs Mx1 and ISG15 and of the C–X–C motif chemokine ligand 10 (CXCL10) messenger RNA (mRNA) expression at day 2 and 5 post-infection, with no difference between the placebo and the IFN-treated groups (Fig 4A). The mRNA expression levels of IFN-γ, and the interleukins (ILs) IL-10 and IL-6 were also significantly upregulated in the infected animals at day 5 post-infection. Similar results were obtained for other immune markers analyzed by RT-qPCR in the lungs (S4 Fig), nasal turbinates (S5 Fig) and spleen (S6 Fig). We also measured the protein levels of chemokine and cytokines either in the lungs or plasma using a commercial enzyme-linked immunosorbent assay (ELISA) directed against hamster IL-6 or a custom-developed hamster multiplex assay. Compared to non-infected animals, we detected an upregulation of CXCL10 and IL-10 protein levels in the lung of all infected groups, with no difference between the placebo and the IFN-treated groups (Fig 4B). We detected a reduction of lung IL-1β levels in IFN-treated groups compared to placebo. Interestingly, lung IL-6 protein level and plasmatic chemokine ligand 2 (CCL2) and tumor necrosis factor-α (TNF-α) protein levels were upregulated in the IFN-late group, compared to the IFN-pre and IFN-early groups (Fig 4B and 4C).

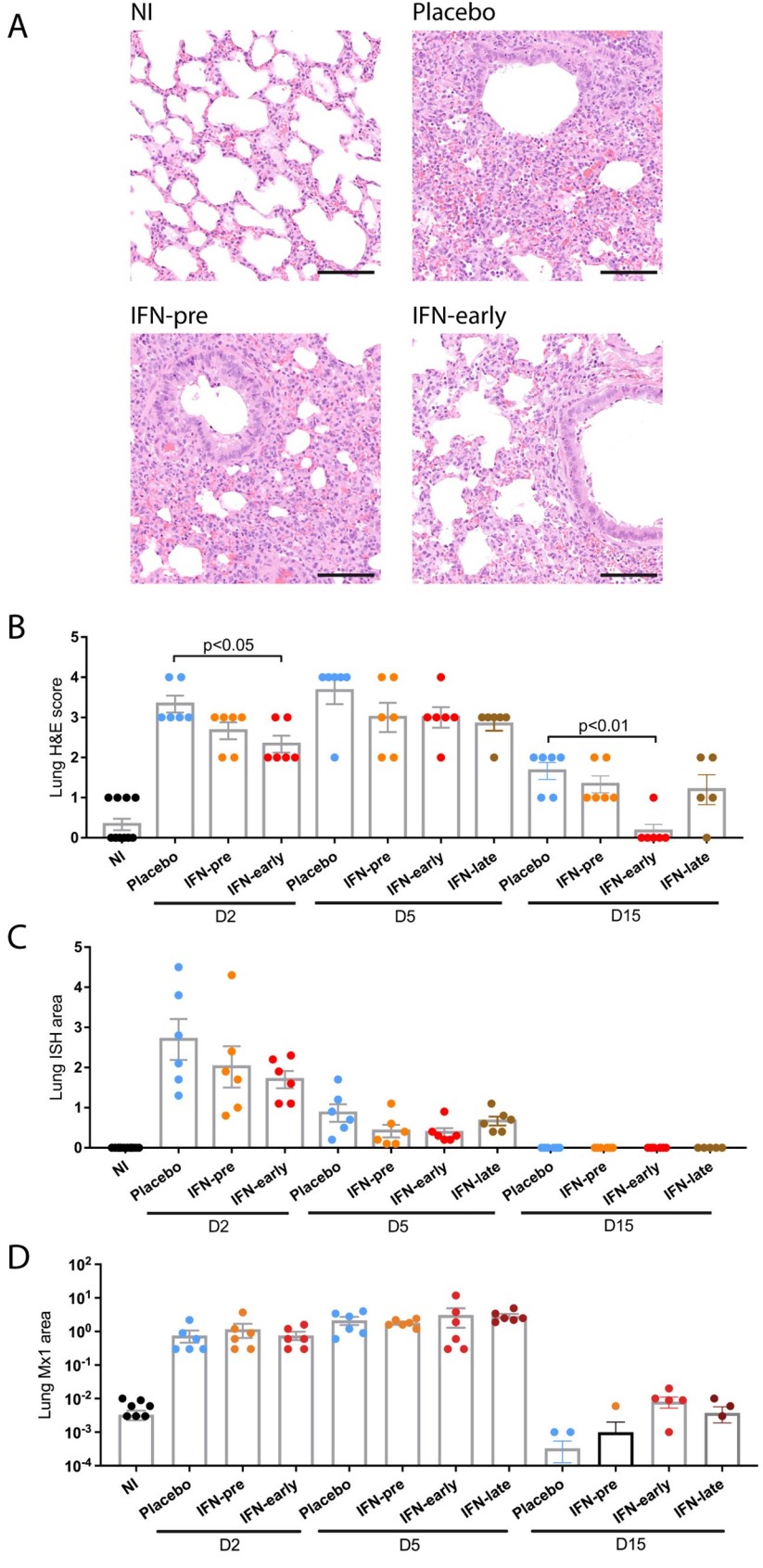

**Fig 3. Histopathological analysis of the impact of type I IFN-α treatment. (A)** Representative pictures were selected to display the pathology from haematoxylin and eosin (H&E) stained lung section from animals at day 2 post infection. Scale bar: 100μm. **(B)** Severity of lung pathology based on lesional scores evaluated from haematoxylin and eosin (H&E) stained lung section. Statistical analysis: Mann-Whitney test. **(C)** Quantification of percent lung area positive for viral RNA in lung sections stained with RNAScope *in situ* hybridization (ISH). Statistical analysis: one-way ANOVA with Tukey's multiple comparisons test. **(D)** Quantification of percent lung area positive for Mx1 protein detected by immunohistochemistry (IHC). Statistical analysis: one-way ANOVA with Tukey's multiple comparisons test. D2: day 2 post infection; D5: day 5 post infection; D15: day 15 post infection. Results are expressed as means ± SEM.

## Discussion

In this study, we assessed the *in vivo* prophylactic and therapeutic efficacy of type I IFN treatment against SARS-CoV-2 infection in the hamster model. Our study demonstrates that type I IFN treatment is beneficial when administered prophylactically or one day post-infection. We observed a significant protection from weight loss in the IFN-pre and in the IFN-early groups, which was associated with a modest reduction of lung viral titers. Interestingly, prophylactic intranasal administration of type I IFN has recently been shown to significantly reduce SARS-CoV-2 replication in hamsters [22]. However, this study did not provide evidence of a protection from weight loss, likely because the inoculum dose of $10^2$ plaque forming units SARS-CoV-2 was not associated with overt clinical signs [22]. We chose a high SARS-CoV-2 inoculum dose of $10^4$ $TCID_{50}$ to induce clinical signs and significant weight loss, in an effort to model patients requiring therapy. The modest reduction in lung viral titers observed upon prophylactic type I IFN treatment in our study is unlikely due to the dose of type I IFN, $10^5$ IU in our study, versus $2.10^5$ IU in [22], nor to the treatment frequency as we observed sustained Mx1 protein expression over 48 hours in the non-infected IFN treated hamsters (Fig 1D). By contrast, we hypothesize that the modest reduction in lung viral titers observed upon prophylactic type I IFN treatment in our study could be due to the fact that we used a high viral inoculum. Interestingly, a similar observation was made in MERS-CoV infected mice: a significant reduction of viral titers was observed upon early treatment with interferon-β in mice inoculated with 750 pfu MERS-CoV [16], while no reduction was observed in mice treated prophylactically with interferon-β and infected with $5.10^4$ pfu MERS-CoV [23]. Intranasal treatment with type I IFN at day one post-infection reduced clinical signs as efficiently as prophylactic treatment in SARS-CoV-2 infected hamsters. Similar findings were obtained when SARS-CoV-2-infected mice were treated prophylactically or at 12 hours post-infection with type III interferon or with the synthetic viral RNA analog poly(I:C) [24,25]. Altogether, these studies demonstrate that stimulation of the antiviral innate immune response before infection or at the very early stage of infection inhibits SARS-CoV-2 replication and pathogenesis, as expected given the high level of SARS-CoV-2 sensitivity to prophylactic type I and type III IFN treatment observed in cell culture [2–4]. By contrast, our study provides the first evidence that administration of type I IFN as soon as the animals exhibited the first clinical signs, corresponding to weight loss, three days post-infection, was not associated with any change in clinical signs compared to placebo treated hamsters. This study thus does not support the use of intranasal type I IFN as a therapeutic in patients with COVID-19 symptoms.

In comparison to humans, virus replication and lung pathology progress much faster in hamsters, which have a peak of virus replication in the lungs at day 2–3 and strongest expression of clinical signs at day 6–7 [20]. Treatment at day 3 post-infection thus corresponds to a "late" time point for treatment initiation in hamsters. We detected an upregulation of IL-6, CCL2 and TNF-α protein levels in the IFN-late group, compared to the IFN-pre and IFN-early groups. However, this did not result in enhanced pathology compared to the placebo group.

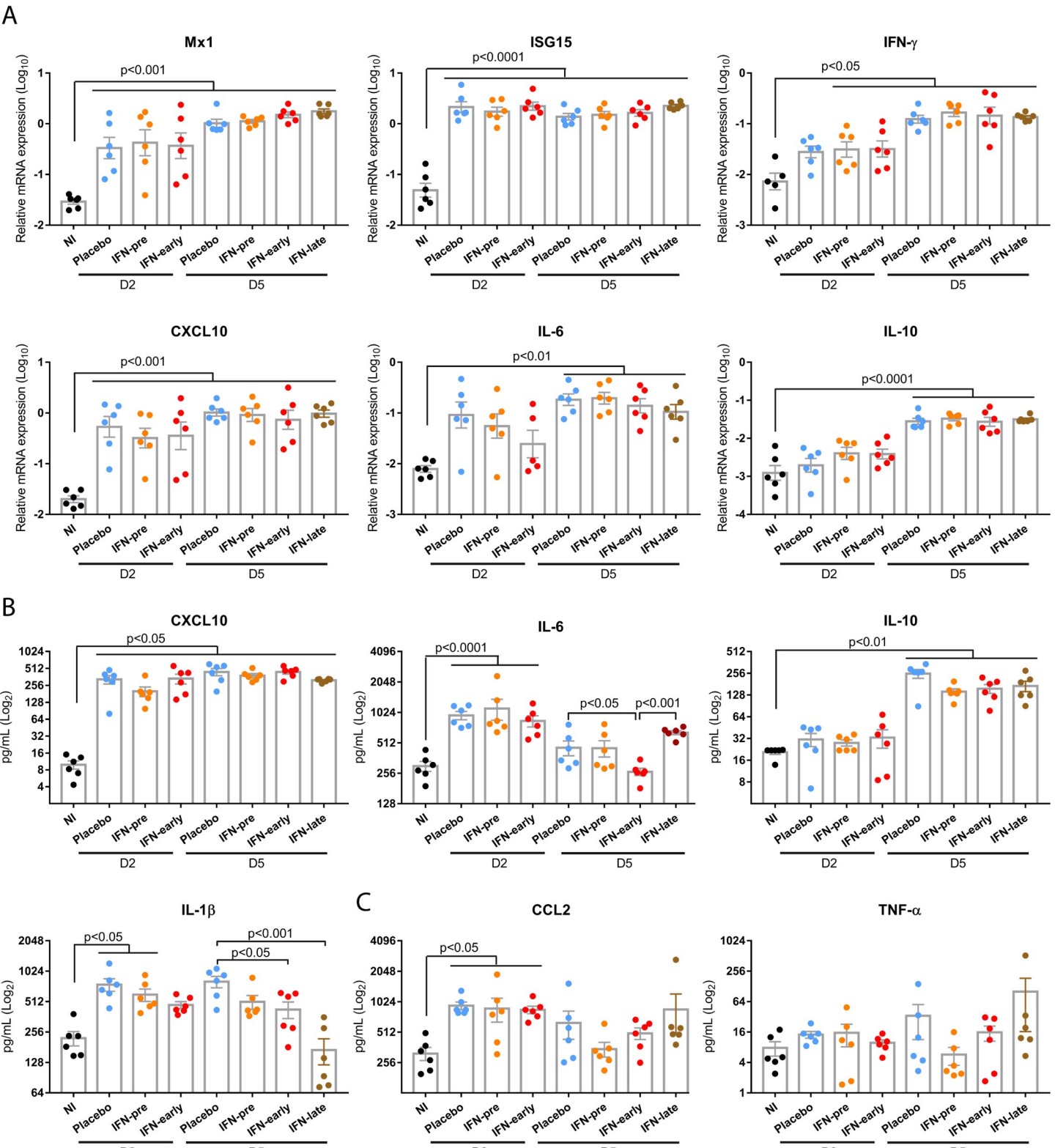

**Fig 4. Impact of type I IFN-α treatment on the immune response to SARS-CoV-2. (A)** Lung transcripts levels of Mx1, ISG15, IFN-γ, CXCL10, IL-6 and IL-10 relative to the housekeeping genes RPL18 and RPS6KB1 determined by RT-qPCR. **(B)** Lung protein levels for CXCL10, IL-6, IL-10 and IL-1β protein levels determined by ELISA or a multiplex assay. **(C)** Plasmatic protein levels for CCL2 and TNF-α determined by a multiplex assay. D2: day 2 post infection; D5: day 5 post infection; D15: day 15 post infection. Results are expressed as means ± SEM. Statistical analysis: one-way ANOVA with Tukey's multiple comparisons test.

Our results therefore indicate that type I IFN treatment at late time-points is unlikely to be associated with deleterious immunopathology exacerbation mechanisms, which were observed in SARS-CoV-1 and MERS-CoV infected mice and feared in SARS-CoV-2-infected humans [13,15,16]. A contribution of type I IFNs to SARS-CoV-2 lung pathology has been suggested from work in IFNAR knockout mice transduced with adenovirus expressing human ACE2 and in STAT2 knockout hamsters [24,26,27]. However, heightened viral loads were also observed in IFNAR knockout mice and STAT2 knockout hamsters, illustrating the fact that type I IFNs can have beneficial and deleterious effects most likely depending on the stage of SARS-CoV-2 infection [24,27]. SARS-CoV-2 expresses a broad array of type I IFN signaling antagonists that likely account for the low sensitivity observed in post-infection type I IFN treatments in cell culture (S7 Fig), and for the lack of beneficial effects observed in this study when type I IFN treatment was initiated at the onset of symptoms three days post-infection [4,28–31].

Even though SARS-CoV-2 expresses a broad array of type I IFN signaling antagonists, we detected a significant upregulation of ISGs expression in the respiratory tract of SARS-CoV-2 infected hamsters, similarly to what has been described in COVID-19 patients [32]. Interestingly, ISG expression in the respiratory tract was not further increased by IFN treatment, as previously observed in MERS-CoV infected mice treated with IFN-beta at 2 days post-infection [16]. This result suggests that ISG levels had reached their maximal expression in response to virus-induced endogenous type I and type III IFNs production and could not be further augmented following exogenous type I IFN administration.

Our study demonstrates that the timing of the type I IFN treatment is critical for its efficacy in a preclinical model of severe SARS-CoV-2 infection. Results from the SOLIDARITY clinical trial showed no benefit of subcutaneous interferon-β-1a injection, while a phase-two clinical trial provided evidence of some benefits of inhaled interferon-β-1a in COVID-19 patients [12,33]. The route of type I IFN administration was not the sole difference between these trials, as the patients treated in the SOLIDARITY trial were on average at a more severe stage of the disease. Our findings support the hypothesis that type I IFN treatment may only be beneficial in patients with low viral load or with mild symptoms at the early stages of the disease, while it likely does not provide any benefit in COVID-19 patients requiring hospitalization [31,34].

## Materials and methods

### Ethics statement

The animal experimentation protocols complied with the regulation 2010/63/CE of the European Parliament and of the council of 22 September 2010 on the protection of animals used for scientific purposes. These experiments were approved by the Anses/ENVA/UPEC ethic committee and the French Ministry of Research (Apafis n˚24818–2020032710416319).

### Animals and interferon-α treatment

We purchased (n = 78) eight week-old female Syrian golden hamsters (*Mesocricetus auratus*, strain RjHan:AURA) from Janviers's breeding Center (Le Genest, St Isle, France) and housed them in an animal-biosafety level 3 (A-BSL3), with *ad libidum* access to water and food. Animals weighing on average 97 grams at 1 day before infection (dbi) were randomly assigned to five groups: 12 non-treated non-infected animals (NI), 18 placebo-treated infected animals (Placebo), 18 infected animals interferon (IFN)-α-treated at 16 hours before infection (hbi), 1 day post-infection (dpi) and 3 dpi (IFN-pre), 18 infected animals IFN-treated at 1 and 3 dpi (IFN-early), and 12 infected animals IFN-treated at 3 and 5 dpi (IFN-late).

Human equivalent dosing in hamsters was calculated based on the dosing for subcutaneous interferon-β-1a ($12 \times 10^6$IU) and inhaled interferon-β-1a ($6 \times 10^6$IU) in clinical trials in COVID-19 patients, corresponding to an average of $1.5 \times 10^5$IU/kg [12,33]. The human dose was multiplied by 7.4 in order to get the hamster equivalent dose of $10^6$IU/kg or $10^5$IU/hamster weighing on average 100g, based on body surface area [19].

At day 1, 3 and 5, all animals were anesthetized with isoflurane and treated by the intranasal route either with 150μL (75μL in each nostril) PBS (Placebo) or with 150μl (75μL in each nostril) PBS containing $10^5$ IU of recombinant universal IFN-α (IFN) (Hu-IFN-αA/D[Bg/II], pbl assay science, Piscataway, NJ). Animals from group IFN-pre were also anesthetized and IFN-treated 1 day prior to infection. In a preliminary experiment designed to test the efficacy of recombinant universal IFN-α in Syrian hamsters, (n = 6) eight week-old female Syrian golden hamsters (*Mesocricetus auratus*, strain RjHan:AURA), purchased from Janviers's breeding Center (Le Genest, St Isle, France) were treated by the intranasal route with 150μl (75μL in each nostril) PBS containing $10^5$ IU IFN. At day 1 post-treatment the animals were euthanized to harvest tissues for gene expression analyses. In a third experiment, (n = 20) eight week-old (*Mesocricetus auratus*, strain RjHan:AURA), also purchased from Janviers's breeding Center (Le Genest, St Isle, France), were either treated by the intranasal route with 150μL (75μL in each nostril) PBS (Placebo) or with 150μl (75μL in each nostril) PBS containing $10^5$ IU IFN, or with 150μl (75μL in each nostril) PBS containing $7.10^5$ IU IFN. Tissues were harvested either at day 1 or day 2 post-treatment, for gene expression and protein levels analysis.

## Virus and experimental infection

SARS-CoV-2 strain UCN1 was amplified as described previously and used at passage 2 [35]. The viral stock was sequenced by Eurofins Genomics (Ebersberg, Germany) using the Illumina deep sequencing Eurofins Genomics Covid Pipeline v.0.1. Sequence analysis revealed that the virus had an intact spike cleavage site. Animals were anesthetized with isoflurane and intranasally inoculated with $10^4$ TCID$_{50}$ units of UCN1 SARS-CoV-2 strain split in 20μL in each nostril. Non-infected animals received the equivalent amount of PBS. Animals were weighted daily from 1 dbi to 15 dpi. Oro-pharyngeal swabs were performed daily from 1 dpi to 6 dpi and at 8, 10 and 12 dpi. Six animals from groups Placebo, IFN-pre and IFN-early were anesthetized and euthanized by exsanguination at 2 dpi and then necropsied. Six animals from each group were also necropsied at 5 dpi. All remaining animals were necropsied at 15 dpi. For each necropsied animal, the following samples were collected: EDTA whole blood, lungs, spleen and nasal turbinates. Organs were either stored frozen at -80˚C in TRIzol reagent (Invitrogen, Carlsbad, CA) or Dulbecco's Modified Eagle medium (DMEM) containing penicillin and streptomycin, or stored in 10% neutral formalin.

## Histology

Samples from lung, upper trachea and larynx were fixed by immersion in 10% neutral-buffered formalin and processed routinely into paraffin wax. 4 μm sections were cut and stained with haematoxylin and eosin (H&E). In addition, samples were stained using the RNAscope in-situ hybridization (ISH) technique to identify the SARS-CoV-2 virus RNA, as previously described [36]. Briefly, tissues were pre-treated with hydrogen peroxide for 10 minutes (room temperature), target retrieval for 15 mins (98–101˚C) and protease plus for 30 mins (40˚C) (Advanced Cell Diagnostics). A V-nCoV2019-S probe [Cat No. 848561, Advanced Cell Diagnostics] was incubated on the tissues for 2 hours at 40˚C. Amplification of the signal was carried out following the RNAscope protocol using the RNAscope 2.5 HD Detection kit–Red (Advanced Cell Diagnostics). Tissue sections were also stained using immunohistochemistry

(IHC) to detect Mx1 using a monoclonal antibody (Merck Sigma-Aldrich MABF938 clone M143/CL143) at 1:1,000. Tissues were dewaxed before heat-induced epitope retrieval was performed using Leica ER1 (pH 6.0) solution for 20 minutes on the Leica Bond RXm automatic stainer. Tissues were treated with hydrogen peroxide (5 mins) and a universal blocker (Superblock TBS blocking buffer, Thermo Scientific; 15 mins) before incubating the antibody for 15 mins. The Leica Bond Polymer Refine detection kit was used for visualisation and counterstaining. Tissue slides were scanned with a Hamamatsu Nanozoomer S360 scanner, visualized with NDP.view2 software and examined by a veterinary pathologist blind to the experimental conditions. A semi-quantitative scoring system was applied to evaluate the lung lesions induced by SARS-CoV-2 as follows: 0 = no lesions; 1 = occasional airway epithelial degeneration and alveolar wall/space infiltration affecting up to 5% of the slide; 2 = mild airways epithelial cell degeneration and alveolar wall and space infiltration, multifocal affecting up to 25% of the slide with presence of peribronchiolar and perivascular cuffs, normally incomplete; 3 = moderate presence of airway epithelial cells degeneration and alveolar wall and space infiltration, multifocal affecting up to 50% of the slide with presence of abundant peribronchiolar and perivascular cuffs; 4 = marked presence of airway epithelial cells degeneration and alveolar wall and space infiltration, multifocal affecting more than 50% of the slide with presence of abundant large peribronchiolar and perivascular cuffs. Digital image analysis Nikon-NIS-Ar software (version 4.30.01) was used to calculate the percentage of area with lesion and with positive staining in ISH and IHC slides.

## Hematology

For each necropsied animal, a complete blood count was performed within 15 minutes of sampling on a ProCyte Dx analyser (IDEXX laboratories, Westbrook, ME). Blood films were also performed, air-dried, stained with May-Grünwlad-Giemsa stain, fixed and coverslip-mounted. They were examined by a board-certified veterinary pathologist, blinded to the experimental conditions, to estimate the leukocyte differential count. The percentages of neutrophils, lymphocytes, monocytes, eosinophils and basophils were estimated from 100 cells. Samples with blood clots were excluded from the hematological analysis.

## RNA extraction from tissue samples and cDNA synthesis

For each organ, 30 mg portions of tissue were placed in tubes with beads (Precellys lysis kit; Stretton Scientific, Ltd., Stretton, United Kingdom) filled with 500 μL of TRIzol reagent (Invitrogen, Carlsbad, CA) and mixed for 5 s at 6,000 rpm three times in a bead beater (Precellys 24; Bertin Technologies, Montigny-le-Bretonneux, France). After TRIzol extraction, the aqueous phase was transferred to 96-well plate and processed according to the manufacturer's instructions (NucleoMag RNA; Macherey-Nagel GmbH & Co, Germany) using a KingFisher automated platform (Thermo Fisher Scientific, Inc., Ontario, Canada). cDNA was synthesized by reverse transcription of 500 ng of total RNA using both oligo(dT)18 (0.25 μg) and random hexamer (0.1 μg) and a RevertAid first-strand cDNA synthesis kit (Invitrogen, Thermo Fisher Scientific) according to the manufacturer's instructions.

## Quantitative PCR from tissue samples

Quantitative PCR for the analysis of host genes expression was performed in 96-well plates in a final volume of 10 μL on a LightCycler 96 (Roche, Mannheim, Germany). Mixes were prepared according to the manufacturer's instructions (QuantiFast SYBR green PCR; Qiagen) with 1 μL of 1:2 diluted cDNA and a final 1 μM concentration of each primer (S1 Table).

Relative quantification was carried out by using the $2^{-\Delta CT}$ method, and the geometric means of two housekeeping genes (RPL18 and RPS6KB1).

## Quantitative PCR from oropharyngeal swabs

Viral RNA was extracted from 160 μL of oro-pharyngeal swabs stored in 500μL of DMEM with antibiotics. RNA extraction was performed by using Qiagen Viral RNA mini kit according to the manufacturer's instructions (Qiagen, Les Ulis, France), with the addition of 15μL Triton X-100 (MP Biomedicals, Illkirch, France) to 560 μL of AVL Lysis buffer for each sample. TaqMan RT-qPCR was performed according to the manufacturer's instruction (Quanti-Tect Probe RT-PCR; Qiagen) using primers targeting the envelope protein gene (E gene) and a previously described protocol [37]. Absolute quantification was performed using a standard curve based on six 10-fold dilutions of a quantitative Synthetic RNA from SARS-CoV-2 (BEI Resources: Catalog No. NR-52358).

## IL-6 ELISA from lung samples

150 mg portions of lung tissue were placed in tubes with beads (Precellys lysis kit; Stretton Scientific, Ltd., Stretton, United Kingdom) filled with 1 mL of Dulbecco's Modified Eagle medium (DMEM) and mixed for 5 s at 6,000 rpm three times in a bead beater (Precellys 24; Bertin Technologies, Montigny-le-Bretonneux, France). Supernatant was collected after a brief centrifugation and a commercial hamster IL-6 double antibody enzyme-linked immunosorbent assay kit was performed according to the manufacturer's instructions (ELISA Genie, Dublin, Ireland).

## Cytokine/Chemokine quantitation using a multiplex assay

150 mg portions of lung tissue were placed in tubes with beads (Precellys lysis kit; Stretton Scientific, Ltd., Stretton, United Kingdom) filled with 1 mL of Dulbecco's Modified Eagle medium (DMEM) containing Triton X100 (1% v/v to inactivate the virus) and a protease inhibitor cocktail (Sigma). Tubes were mixed for 5 s at 6,000 rpm three times in a bead beater (Precellys 24; Bertin Technologies, Montigny-le-Bretonneux, France). Plasma samples also contained Triton X100 (1% v/v to inactivate the virus) and a protease inhibitor cocktail (Sigma). The custom Syrian Hamster Panel was developed by Merck-Millipore under the reference number SPRCUS1249, using previously identified cross-reactivity with the potential to detect hamster proteins from pre-developed commercial assays for rat (RECYTMAG-65K) and feline (FCYTMAG-20K) species, respectively. It was used for the measurement of IL-1β, CCL2, TNFα, CXCL10, and IL-10. The custom Syrian Hamster Milliplex xMAP kit (SPRCUS1249, Merck-Millipore) is available upon request to the corresponding author. Data were recorded on a MagPix instrument using Xponent software (Luminex). Results are expressed as concentration in pg/mL. Among the cytokines and chemokines included in the multiplex assay, we detected a robust signal for CXCL10, IL-10, IL-1β in lung and plasma samples. However, CCL2 and TNF-α were only detected in plasma samples, most likely due to higher signal to noise ratio in plasma samples than in lung samples.

## *In vitro* IFN-α treatment

Vero-E6 cells were grown in DMEM containing 1% antibiotics (penicillin/streptomycin) and 10% fetal bovine serum. Cells were treated with 1000 UI/mL recombinant universal IFN-α (IFN) (Hu-IFN-αA/D[Bg/II], pbl assay science, Piscataway, NJ) for 18 hours prior to infection and were subsequently infected with SARS-CoV-2 at a multiplicity of infection (MOI) of $10^{-3}$.

Infections were carried out in DMEM containing 1% antibiotics (penicillin/streptomycin) and 2% fetal bovine serum. Recombinant universal IFN-α (Hu-IFN-αA/D[Bg/II], pbl assay science, Piscataway, NJ) was added at a final concentration of 1000 UI/mL to the medium 6 hours post-infection for cells treated subsequently to the infection. Culture supernatants were harvested 24 hours post-infection and viral titers were determined by the $TCID_{50}$ method on Vero-E6 cells and calculated by the Spearman & Kärber algorithm.

## SARS-CoV-2 titrations from lung samples

Titrations were performed on 90% confluent Vero-E6 cells in 96-well plates. Viral titers were calculated using the Spearman-Kärber method.

## Statistical analyses

All the data, except weight loss and H&E lesion scoring, were statistically analyzed using one-way analysis of variance (ANOVA) followed by Tukey's post hoc test (GraphPad Prism Software, USA) with the p values indicated corresponding to the results of Tukey's post hoc tests. Weight loss was statistically analyzed using two-way ANOVA with Geisser-Greenhouse correction followed by Tukey's multiple comparisons test (GraphPad Prism Software, USA) with the p values indicated for the weight loss corresponding to the results of Tukey's post hoc tests. H&E lesion scores were statistically analyzed using the Mann-Whitney test.

## Supporting information

**S1 Table. List of primers used in this study.**
(DOCX)

**S1 Fig. Subgenomic viral RNA in nasal turbinates.** Nasal turbinates were harvested at day 2 post-infection (D2) or day 5 post-infection (D5). Viral sgRNA levels relative to the housekeeping genes RPL18 and RPS6KB1 were determined by RT-qPCR. Results are expressed as means ± SEM. Statistical analysis: one-way ANOVA with Tukey's multiple comparisons test.
(TIF)

**S2 Fig. Histological analysis of the impact of IFN-α treatments.** Representative pictures were selected to display the pathology from haematoxylin and eosin (H&E) stained lung section, viral RNA in lung sections stained with RNAScope *in situ* hybridization (ISH) and Mx1 protein detected by immunohistochemistry (IHC). D2: day 2 post infection; D5: day 5 post infection; D15: day 15 post infection. Scale bar: 100μm.
(TIF)

**S3 Fig. Type I IFN-α treatment does not prevent lymphocytopenia.** A complete blood count analysis was performed as described in the methods section. D2: day 2 post infection; D5: day 5 post infection. Results are expressed as means ± SEM. Statistical analysis: one-way ANOVA with Tukey's multiple comparisons test.
(TIF)

**S4 Fig. Impact of type I IFN-α treatment on the immune response to SARS-CoV-2. in the lung.** Lung transcripts levels of IL-1β, IL-12, IFN-α7, OAS3, Mx2, CCL5 relative to the housekeeping genes RPL18 and RPS6KB1 determined by RT-qPCR. D2: day 2 post infection; D5: day 5 post infection; D15: day 15 post infection. Results are expressed as means ± SEM. Statistical analysis: one-way ANOVA with Tukey's multiple comparisons test.
(TIF)

**S5 Fig. Impact of IFN-α treatment on the immune response to SARS-CoV-2 in the nasal turbinates.** Nasal turbinates transcripts levels of Mx1, Mx2, ISG15, IFN-α7, CXCL10, CCL5, IL-1β, IL-6, IL-10, IFN-γ and IL-12 relative to the housekeeping genes RPL18 and RPS6KB1 determined by RT-qPCR. D2: day 2 post infection; D5: day 5 post infection; D15: day 15 post infection. Results are expressed as means ± SEM. Statistical analysis: one-way ANOVA with Tukey's multiple comparisons test.
(TIF)

**S6 Fig. Impact of IFN-α treatment on the immune response to SARS-CoV-2. in the spleen.** Spleen transcripts levels of Mx1, Mx2, ISG15, IFN-α7, CXCL10, CCL5, IL-1β, IL-6, IL-10, IFN-γ and IL-12 relative to the housekeeping genes RPL18 and RPS6KB1 determined by RT-qPCR. D2: day 2 post infection; D5: day 5 post infection; D15: day 15 post infection. Results are expressed as means ± SEM. Statistical analysis: one-way ANOVA with Tukey's multiple comparisons test.
(TIF)

**S7 Fig. Effect of IFN-α treatment on SARS-CoV-2 replication in cell culture.** Vero-E6 cells were treated either with placebo or with $10^3$ UI/mL recombinant universal IFN-α 18 hours prior to infection (IFN-pre) or 6 hours post infection (IFN-post). Equivalent volume of PBS was used as a negative control. Viral titers were determined by $TCID_{50}$ from supernatants collected 24 hours post infection. Each dot represents a technical replicate of a representative experiment performed twice. Results are expressed as means ± SEM. Statistical analysis: one-way ANOVA with Tukey's multiple comparisons test.
(TIF)

## Acknowledgments

We thank Meriadeg Ar Gouilh and Astrid Vabret (University Hospital of Caen, Normandie Université, Caen, France) for granting access to the SARS-CoV-2 UCN1 strain and Daniel Gonzalez-Dunia for proofreading the manuscript. We thank Vanessa Bastid, Mélanie Badré-Biarnais, Jean-Marc Boucher, Anouck Labadie, Carine Peytavin de Garam, Jonathan Rieder and Jean-Luc Schereffer for their investment in virological and serological analyses; Valère Brogat, Sébastien Kempff and Estelle Litaize for animal care and experimentation (Nancy laboratory for rabies and wildlife, ANSES, Lyssavirus Unit, Malzéville, France).

## Author Contributions

**Conceptualization:** Pierre Bessière, Gilles Foucras, Elodie Monchatre-Leroy, Romain Volmer.

**Formal analysis:** Pierre Bessière, Gilles Foucras, Romain Volmer.

**Funding acquisition:** Pierre Bessière, Gilles Foucras, Elodie Monchatre-Leroy, Romain Volmer.

**Investigation:** Pierre Bessière, Marine Wasniewski, Evelyne Picard-Meyer, Alexandre Servat, Thomas Figueroa, Charlotte Foret-Lucas, Amelia Coggon, Sandrine Lesellier, Frank Boué, Nathan Cebron, Blandine Gausserès, Catherine Trumel, Gilles Foucras, Francisco J. Salguero, Elodie Monchatre-Leroy, Romain Volmer.

**Methodology:** Pierre Bessière, Marine Wasniewski, Evelyne Picard-Meyer, Alexandre Servat, Thomas Figueroa, Charlotte Foret-Lucas, Amelia Coggon, Sandrine Lesellier, Frank Boué, Nathan Cebron, Blandine Gausserès, Catherine Trumel, Gilles Foucras, Francisco J. Salguero, Elodie Monchatre-Leroy, Romain Volmer.

**Project administration:** Romain Volmer.

**Supervision:** Gilles Foucras, Elodie Monchatre-Leroy, Romain Volmer.

**Writing – original draft:** Pierre Bessière, Francisco J. Salguero, Romain Volmer.

**Writing – review & editing:** Pierre Bessière, Gilles Foucras, Francisco J. Salguero, Romain Volmer.

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
