## [Decision Letter · Decision Letter 0]

26 Mar 2021

Dear Dr. Volmer,

Thank you very much for submitting your manuscript "Intranasal type I interferon treatment is beneficial only when administered before clinical signs onset in the SARS-CoV-2 hamster model" for consideration at PLOS Pathogens. As with all papers reviewed by the journal, your manuscript was reviewed by members of the editorial board and by several independent reviewers. In light of the reviews (below this email), we would like to invite the resubmission of a significantly-revised version that takes into account the reviewers' comments.

Note that the reviews were mixed with one recommending rejection in large part based on novelty due to the published paper by the TenOever laboratory. The other reviewers and I feel there are some additional interesting (although not completely clear) results that could make this paper a sound addition to the SARS-CoV-2 field. In the revision, you will need to address the following: (a) make sure key experiments have independent biological repeats at separate times; (b) provide greater information and methods on the cytokine assay; (c) perform additional ISH and histological analysis as suggested by the Reviewers and (d) provide more discussion as to the lack of antiviral effect of IFN in some of the experiments and how this relates to prior studies.

We cannot make any decision about publication until we have seen the revised manuscript and your response to the reviewers' comments. Your revised manuscript is also likely to be sent to reviewers for further evaluation.

Sincerely,

Michael Diamond

Section Editor

PLOS Pathogens

Kasturi Haldar

Editor-in-Chief

PLOS Pathogens

orcid.org/0000-0001-5065-158X

Michael Malim

Editor-in-Chief

PLOS Pathogens

orcid.org/0000-0002-7699-2064

Reviewer's Responses to Questions

**Part I - Summary**

Reviewer #1: Bessière et al evaluated the potential therapeutic effects of high dose type I interferon treatment against SARS-CoV-2 infection in the Syrian hamster model. One day before or after infection, high dose IFN treatment had a modest effect on weight loss. Starting IFN treatment 3 dpi, no effect was observed. Surprisingly, pre-treatment with type I IFN had no effect on infectious virus titer and viral RNA load. In addition, no significant difference in lung pathology was observed in the pre-treated animals, albeit this may be due to the relatively small number of animals per group. Initiating IFN treatment one day after infection had significant effects on lung pathology 2 dpi, and reduced viral RNA load and infectious virus titer 5 dpi. No effect on weight loss, inflammation or virus titer was observed in hamsters treated with IFN starting 3 dpi. Combined these data are rather surprising and it poses the question, why does type I IFN not protect hamsters from SARS-CoV-2 challenge. The authors demonstrate increased Mx2 gene expression following treatment, suggesting that the universal type I IFN is able to induce an antiviral immune response. Another possibility is that the antiviral immune response is not effective against SARS-CoV-2, or the antiviral immune response is not induced in the target cell of SARS-CoV-2. To distinguish between the two possibilities, the authors should perform their Mx1 RNA-ISH staining on interferon treated animals without SARS-CoV-2 challenge. This allows the authors to provide additional insight as to why type I IFN provides minimal protection and compare the antiviral response between treatment and actual infection. Finally, the authors present some intriguing cytokine and chemokine protein data by ELISA and multiplex array on hamster lung homogenates and sera. If these were available to the general scientific community, this would be a major advance for the field. However, minimal details are provided in terms of product number, antibody clones etc. A search for Syrian Hamster cytokines MilliPlex xMAP assay on the company website yields no hits. AssayGenie appears to have several ELISA kits including IL-6, IL-1b etc., but there is no mention of CXCL10 or CCL2. Additional information must be provided to allow others to make use of these reagents and assays.

Major comments:

• All experiments appear to have been done in one large experiment. The authors should consider repeating key findings in a second experiment.

• Details on the ELISA or bead-array kits are missing. What kind of

• RNA-ISH using Mx1 on IFN treated animals will provide important insight into the reason for the lack of a robust effect of type I IFN administration prior to virus challenge.

• Please discuss the surprising lack of IFN effect on virus titers and inflammation in this model.

Minor:

• Type in the methods section on the hamsters. It appears they are 78 weeks old….probably 7-8 weeks old.

• Provide some more details in the “main text” such as the source and dose the type I IFN used. This can be found in the materials and methods section, but should be referenced elsewhere.

Reviewer #2: ‘Intranasal type I interferon treatment is beneficial only when administered before clinical signs onset in the SARS-CoV-2 hamster model’ by Bessière et al. examines the impact of intranasal delivery of universal IFNa to SARS-CoV-2 infected hamsters in either prophylactic or therapeutic treatment models. The article is well written and contains appropriate citations of the recent literature. While much of the data from this manuscript confirms previously published works, new details concerning the timing of treatment are revealed in this manuscript and are important to the field. The most dramatic differences in treatment result appear to be in lung histopathology; it would greatly improve the manuscript if this data was expanded upon with high magnification images and characterization of immune cell infiltrates. Overall, this paper demonstrates modest efficacy of type I IFN treatment in preventing the most serious signs of disease including reduced viral titer in the lung, improved lung histopathology and reduction in some cytokine/chemokine levels. However given the compressed treatment window in small animal models, these results are still significant and applicable to COVID-19 patient treatment.

Reviewer #3: Bessiere et al. use the hamster model of SARS-CoV-2 infection to test the efficacy of IFNa treatment on infection on outcome regarding lung viral loads and pathology. For us it not clear where the novelty of the results are, in particular as very similar data have been presented earlier in the same model (Hoagland et al. Immunity 2021) yet in much greater detail and trialing many more conditions (dose response, full DEG analysis). Furthermore, the authors fail to provide strong evidence for a a clear benefit from an proposed IFN regimen. Obviously hamsters constitute a very stringent challenge model, however, it had been demonstrated before that efficient therapeutic interventions for SARS-COV-2 are in principle possible in hamsters (e.g. Kaptein et al. PNAS 2020, in that case using RdRP inhibitors) and hence a --if at all-- 1log reduction in viral loads and almost no reduction in any of the measured cytokines may not be considered an incremental gain in knowledge towards promising treatment options.

**Part II – Major Issues: Key Experiments Required for Acceptance**

Reviewer #1: Key findings should be repeated in a second cohort of animals

RNA-ISH in Mx1 in IFN treated animals to demonstrate the location and intensity of the antiviral host response.

Reviewer #2: The most significant and consistent results showing an effect of IFN treatment were in the ‘early’ treatment group. Do the authors think prophylactic treatment would have been more effective if daily dosing was used instead of every 2 day dosing? How long does ISG stimulation last after treatment? Prophylactic treatment results are not strong as the Hoagland et al, Immunity paper.

Figure 2A – Figure 2 shows the strongest protection data in the manuscript and more detail to explain these results is key to demonstrating the utility of IFN treatment. Why are the lung sections shown from D5 post infection when there are not significant differences in histology score? Additional, high magnification, histopathology images should be included.

The authors should discuss why titer changes are observed only in the lower airway when strong ISG induction was observed in nasal turbinates.

Reviewer #3: Dose response for (i) virus inoculum and (ii) interferon for infection outcome.

The quantification of lung areas/volumes affected may need more vigorous means for quantification to tease out differences, if at all.

**Part III – Minor Issues: Editorial and Data Presentation Modifications**

Reviewer #1: (No Response)

Reviewer #2: Figure 1B – Is weight loss significantly different between IFN treated and sham treated groups of hamsters?

It would be very helpful if the titer data in figure 1C was labeled as being from oralpharangeal swabs and D/E being from the lung.

Figure 2B – significance notations are confusing and don’t seem to match the text. Please clarify if all treatment conditions are significant or if only IFN-early led treatment to significant reductions in histopathology.

Reviewer #3: The luminex assays for hamster cytokines are not described but may be of interest for he general public.

PLOS authors have the option to publish the peer review history of their article (what does this mean?). If published, this will include your full peer review and any attached files.

Reviewer #1: No

Reviewer #2: No

Reviewer #3: No
---

## [Decision Letter · Decision Letter 1]

28 Jun 2021

Dear Dr. Volmer,

Thank you very much for submitting your manuscript "Intranasal type I interferon treatment is beneficial only when administered before clinical signs onset in the SARS-CoV-2 hamster model" for consideration at PLOS Pathogens. As with all papers reviewed by the journal, your manuscript was reviewed by members of the editorial board and by several independent reviewers. The reviewers appreciated the attention to an important topic. Based on the reviews, we are likely to accept this manuscript for publication, providing that you modify the manuscript according to the review recommendations. There are just a few remaining editorial and data presentation comments that need to be addressed.

Sincerely,

Michael Diamond

Section Editor

PLOS Pathogens

Kasturi Haldar

Editor-in-Chief

PLOS Pathogens

orcid.org/0000-0001-5065-158X

Michael Malim

Editor-in-Chief

PLOS Pathogens

orcid.org/0000-0002-7699-2064

Reviewer Comments (if any, and for reference):

Reviewer's Responses to Questions

**Part I - Summary**

Reviewer #1: The reviewers have analyzed the Mx1 gene and protein expression in hamsters after IFN treatment. This is very nice data and suggests that SARS-CoV-2 can infect and replicate in cells despite the presence of an antiviral immune response. What they have not done is repeat some of the key findings as requested. While I agree with the reviewers that some P-values are unlikely to change, such as the cytokine induction over mock, others certainly can. For example, the virus titer differences (Fig 1) on day 5 are barely significant.

Reviewer #2: ‘Intranasal type I interferon treatment is beneficial only when administered before clinical signs onset in the SARS-CoV-2 hamster model’ by Bessière et al. examines the impact of intranasal delivery of universal IFNa to SARS-CoV-2 infected hamsters in either prophylactic or therapeutic treatment models. The article is well written and contains appropriate citations of the recent literature. While much of the data from this manuscript confirms previously published works, new details concerning the timing of treatment are revealed in this manuscript and are important to the field. The most dramatic differences in treatment result are in reduced weight loss, improved lung histopathology and a reduction in virus titer in the lung. High magnification images and generally improved data presentation have strengthened this manuscript. Overall, this paper demonstrates modest efficacy of type I IFN treatment in preventing the most serious signs of disease including reduced viral titer in the lung, improved lung histopathology and reduction in some cytokine/chemokine levels. However, given the compressed treatment window in small animal models, these results are still significant and applicable to COVID-19 patient treatment. Most importantly, the authors do not overstate their findings and accurately present them as part of our growing understanding of how to treat COVID-19 patients.

Reviewer #3: I may have to excuse for any possible oversight, maybe partially due to a lack of obvious mark-ups in the revised version of the manuscript available to me, however, it is not very clear to what extent and where the authors provide significant new insight in the revised version of the manuscript. The impression prevails that little efforts was made to consolidate previous claims. As a minimal requirement, the wording in title (now claiming a true benefit from IFN treatment) may need to be tempered. E.g. from “is beneficial” to “might confer some benefit” or similar. Here also stating “lack of therapeutic benefit of intranalsal IFN” may be appropriate and as such an important finding.

**Part II – Major Issues: Key Experiments Required for Acceptance**

Reviewer #1: (No Response)

Reviewer #2: My questions and concerns have been adequately addressed in this revised manuscript.

Reviewer #3: As mentioned before, the importance of the rather small differences regarding any of the parameters measured in infected hamsters (weight evolution, virus loads, RT-qPCR, histology score, cytokine) remains of concern. With this regard, I convene with the Editor and Reviewer #1 that consolidation of data in independent biological repeats may be required, at least for some selected key data; I have the impression that all data on infection outcome were generated from a single batch of animals. I understand the ethical consideration that aims to reduce animal suffering. However, if the authors claim (see line 180-181) that their study has relevance to change clinical practice, independent confirmation should be considered more seriously, including from an ethical point of view either endorsing or not to treat patients that are at high risk of developing severe disease.

**Part III – Minor Issues: Editorial and Data Presentation Modifications**

Reviewer #1: The H&E score in Fig 2 is analyzed with an ANOVA. However, the data appears non-parametric and not normally distributed (day 15, early IFN-a). Please check statistical analysis

Reviewer #2: (No Response)

Reviewer #3: The quantitative scoring for histopathology may be accepted, fully trusting the expert pathologist’s view, however more details on the microscopic parameters and lesions quantified would be appreciated to level up with other studies using the same hamster model. Likewise, regarding my original request for more information on the novel cytokine multiplex assay, I understood the journal’s policy to enforce open science. The multiplex assay used here may be of quite some interest to the field that is in desperate need of new state-of-the-art research tools. More efforts in describing the performance, validation and technical specifications (e.g. list of reagents) of these tools would be very much appreciated.

PLOS authors have the option to publish the peer review history of their article (what does this mean?). If published, this will include your full peer review and any attached files.

Reviewer #1: No

Reviewer #2: No

Reviewer #3: No

Figure Files:

Data Requirements:

Reproducibility:

References:

---

## [Editor Report · Decision Letter 2]

7 Jul 2021

Dear Dr. Volmer,

We are pleased to inform you that your manuscript 'Intranasal type I interferon treatment is beneficial only when administered before clinical signs onset in the SARS-CoV-2 hamster model' has been provisionally accepted for publication in PLOS Pathogens.

Best regards,

Michael S. Diamond

Section Editor

PLOS Pathogens

Michael Diamond

Section Editor

PLOS Pathogens

Kasturi Haldar

Editor-in-Chief

PLOS Pathogens

orcid.org/0000-0001-5065-158X

Michael Malim

Editor-in-Chief

PLOS Pathogens

orcid.org/0000-0002-7699-2064
---

## [Editor Report · Acceptance letter]

4 Aug 2021

Dear Dr. Volmer,

We are delighted to inform you that your manuscript, "Intranasal type I interferon treatment is beneficial only when administered before clinical signs onset in the SARS-CoV-2 hamster model," has been formally accepted for publication in PLOS Pathogens.

Best regards,

Kasturi Haldar

Editor-in-Chief

PLOS Pathogens

orcid.org/0000-0001-5065-158X

Michael Malim

Editor-in-Chief

PLOS Pathogens

orcid.org/0000-0002-7699-2064